# Evolutionary pathways to antibiotic resistance are dependent upon environmental structure and bacterial lifestyle

Alfonso Santos-Lopez[1,2†], Christopher W Marshall[1,2†‡], Michelle R Scribner[1,2], Daniel J Snyder[1,2,3], Vaughn S Cooper[1,2,3]*

[1]Department of Microbiology and Molecular Genetics, University of Pittsburgh, Pittsburgh, United States; [2]Center for Evolutionary Biology and Medicine, University of Pittsburgh, Pittsburgh, United States; [3]Microbial Genome Sequencing Center, University of Pittsburgh, Pittsburgh, United States

**Abstract** Bacterial populations vary in their stress tolerance and population structure depending upon whether growth occurs in well-mixed or structured environments. We hypothesized that evolution in biofilms would generate greater genetic diversity than well-mixed environments and lead to different pathways of antibiotic resistance. We used experimental evolution and whole genome sequencing to test how the biofilm lifestyle influenced the rate, genetic mechanisms, and pleiotropic effects of resistance to ciprofloxacin in *Acinetobacter baumannii* populations. Both evolutionary dynamics and the identities of mutations differed between lifestyle. Planktonic populations experienced selective sweeps of mutations including the primary topoisomerase drug targets, whereas biofilm-adapted populations acquired mutations in regulators of efflux pumps. An overall trade-off between fitness and resistance level emerged, wherein biofilm-adapted clones were less resistant than planktonic but more fit in the absence of drug. However, biofilm populations developed collateral sensitivity to cephalosporins, demonstrating the clinical relevance of lifestyle on the evolution of resistance.

DOI: https://doi.org/10.7554/eLife.47612.001

**\*For correspondence:**
vaughn.cooper@pitt.edu

†These authors contributed equally to this work

**Present address:** ‡Department of Biological Sciences, Marquette University, Milwaukee, United States

**Competing interests:** The authors declare that no competing interests exist.

## Introduction

Antimicrobial resistance (AMR) is one of the main challenges facing modern medicine. The emergence and rapid dissemination of resistant bacteria is decreasing the effectiveness of antibiotics and some estimates suggest as many as 700,000 people die per year due to AMR-related problems (*O'Neill et al., 2016*). AMR, like all phenotypes, is an evolved property, either the ancient product of living amidst other microbial producers of antimicrobials (*Martínez, 2008*), or the recent product of strong selection by human activities for novel resistance-generating mutations (*Ventola, 2015*).

The dominant mode of growth for most microbes is on surfaces, and this biofilm lifestyle is central to AMR (*Høiby et al., 2010*; *Olsen, 2015*; *Ahmed et al., 2018*), especially in chronic infections (*Wolcott et al., 2010*; *Wolcott, 2017*). However, with few exceptions (*Ridenhour et al., 2017*; *Ahmed et al., 2018*; *France et al., 2019*), most of the research on the evolution of AMR has been conducted in well-mixed populations (reviewed in *Hughes and Andersson, 2017*) or on agar plates (*Baym et al., 2016a*), conditions that cannot simulate the effects of biofilms on the evolution of AMR. Consequently, our understanding of how this lifestyle influences the evolution of AMR, whether by different population-genetic dynamics or molecular mechanisms, is limited. One example is that the close proximity of cells in biofilms may facilitate the horizontal transfer and persistence of

**eLife digest** A bacterium known as *Acinetobacter baumannii* causes serious lung infections in people with weakened immune systems. These illnesses are becoming more common largely because *A. baumannii* is increasingly developing resistance to antibiotics.

Inside the airways, individual *A. baumannii* cells can stick together and coat themselves in a slimy substance to form a structure called biofilm, which physically protects bacteria from antibiotics. This may be one of the reasons why it is often harder to treat bacterial infections associated with biofilms.

Another possibility is that bacteria may evolve differently in biofilms compared with cells living independently. For example, *A. baumannii* may colonize several regions of the lungs during an infection, leading to distinct groups of bacteria that experience different conditions and evolve separately. Each population may therefore respond differently to an antibiotic. In contrast, bacteria living independently in a well-mixed population – such as in the bloodstream of their host – would be more likely to all evolve in the same way.

Santos-Lopez, Marshall et al. tested this theory by exposing populations of *A. baumannii* that lived either independently or in biofilms to increasing levels of an antibiotic called ciprofloxacin. The genetic information of these cells was examined as the populations were evolving, and the bacteria were also put in contact with other types of antibiotics.

The analyses revealed that bacteria in well-mixed populations shared the same limited number of mutations: these gave the bacteria high levels of resistance to the antibiotic's primary target, an enzyme involved in DNA processes. The bacteria had also become resistant to other classes of antibiotics.

In contrast, the bacteria in biofilm populations evolved to be more genetically diverse, exhibiting different types of mutations that helped the cells to pump out the drug. These bacteria were less resistant to ciprofloxacin and more sensitive to other types of antibiotics.

Further experiments looked into the fitness of the bacteria – their ability to survive, reproduce and compete with each other. High levels of antibiotic resistance came with lower fitness: biofilm bacteria had evolved to become being fitter than those from well-mixed population. Even in the absence of drugs, these populations were in fact fitter than the original cells.

Overall, understanding how the lifestyles of bacteria affect the way they respond to drugs may help researchers to develop new approaches that limit the spread of antibiotic resistance and improve treatment.

DOI: https://doi.org/10.7554/eLife.47612.002

resistance genes in bacterial populations (*Stalder and Top, 2016*; *Ridenhour et al., 2017*). Less appreciated is the potential for the biofilm lifestyle to influence the evolution of AMR by de novo chromosomal mutations. This emergence of AMR in biofilms is important because: i) the environmental structure of biofilms can increase clonal interference, rendering selection less effective and enhancing genetic diversity (*Habets et al., 2006*; *Traverse et al., 2013*; *Cooper et al., 2014*; *Ellis et al., 2015*; *France et al., 2019*), ii) distinct ecological conditions within the biofilm can favor functionally distinct adaptations to different niches (*Poltak and Cooper, 2011*), iii) the biofilm itself can protect its residents from being exposed to external stresses like antibiotics or host immunity, and weaken selection (*Geisinger and Isberg, 2015*; *Eze et al., 2018*), and iv) slower growth within biofilms can reduce the efficacy of antibiotics that preferentially attack fast-growing cells (*Walters et al., 2003*; *Kirby et al., 2012*). The first two hypotheses would predict more complex evolutionary dynamics within biofilms than in well-mixed environments (*Steenackers et al., 2016*), while the second two predict different rates of evolution, targets of selection, and likely less potent mechanisms of AMR (*Andersson and Hughes, 2014*). Together, these potential factors call into question the conventional wisdom of a tradeoff between fitness and antimicrobial resistance, a relationship that remains to be clearly defined.

Here, we study the evolutionary dynamics and effects of new resistance mutations in the opportunistic nosocomial pathogen *Acinetobacter baumannii*, which is often intrinsically resistant to antibiotics or has been reported to rapidly evolve resistance to them (*Doi et al., 2015*). This pathogen is

categorized as one of the highest threats to patient safety (*Asif et al., 2018*), partly due to its ability to live on inanimate surfaces in biofilms (*Eze et al., 2018*). We experimentally propagated populations of *A. baumannii* exposed either to subinhibitory or increasing concentrations of ciprofloxacin (CIP) over 80 generations in biofilm or planktonic conditions to ascertain whether these lifestyles select for different mechanisms of AMR. Rather than focusing on the genotypes of single isolates, which can limit the scope of an analysis, we conducted whole-population genomic sequencing over time to define the dynamics of adaptation and the fitness of certain resistance alleles compared to others in the experiment. We then identified clones with specific genotypes that we linked to fitness and resistance phenotypes. This approach sheds new light on the ways that pathogens adapt to antibiotics while growing in biofilms and has implications for treatment decisions.

## Results and discussion

### Experimental evolution

Replicate cultures of the susceptible *A. baumannii* strain ATCC 17978 (*Piechaud and Second, 1951*; *Baumann et al., 1968*) were established under planktonic or biofilm conditions in one of three treatments: i) no antibiotics, ii) sub-inhibitory concentration of the antibiotic ciprofloxacin (CIP) and iii) evolutionary rescue (*Bell and Gonzalez, 2009*) in which CIP concentrations were increased every 72 hr from subinhibitory concentrations to four times the minimum inhibitory concentration (MIC) (*Figure 1A*). Before the start of the antibiotic evolution experiment, we propagated the ATCC strain for ten days in planktonic conditions to reduce the influence of adaptation to the laboratory conditions on subsequent comparisons. CIP was chosen because of its clinical importance in treating *A. baumannii* (*Lopes and Amyes, 2013*; *Ardebili et al., 2014*; *Doi et al., 2015*), its ability to penetrate the biofilm matrix (*Tseng et al., 2013*) allowing similar efficacy in well mixed and structured populations (*Kirby et al., 2012*), and because it is not known to stimulate biofilm formation in *A. baumannii* (*Aka and Haji, 2015*). Planktonic populations were serially passaged by daily 1:100 dilution while biofilm populations were propagated using a bead model simulating the biofilm life cycle (*Poltak and Cooper, 2011*; *Traverse et al., 2013*; *Turner et al., 2018*). This model selects for bacteria that attach to a 7 mm polystyrene bead, form a biofilm, and then disperse to colonize a new bead each day. (A video tutorial for this protocol is available at http://evolvingstem.org/see-it-in-action). The transfer population size in biofilm and in planktonic cultures was set to be nearly equivalent at the beginning of the experiment (approximately $1 \times 10^7$ CFU/mL), because population size influences mutation availability and the response to selection (*Salverda et al., 2017*; *Cooper, 2018*). Each day, the population size increases 100-fold during regrowth, generating approximately $10^6$ new mutations per day using a conservative but experimentally justified estimate of the mutation rate (*Lynch et al., 2016*; *Dillon et al., 2017*). Effects of fluoroquinolones like CIP have been shown to increase the mutation rate by an order of magnitude, and some studies suggest biofilm growth may also increase the mutation rate (*Boles and Singh, 2008*; *Long et al., 2016*; *Pribis et al., 2019*). Thus any differences in mutated genes reaching high frequency between treatments are almost certainly the product of selection and not a lack of mutation availability, though early-arising or more probable beneficial mutations could sweep and limit invasion of selectively equivalent mutations in different genes (*Khan et al., 2011*; *Flynn et al., 2013*; *Kryazhimskiy et al., 2014*). The mutational dynamics of three lineages from each treatment were tracked by whole-population genomic sequencing (*Figure 1A*). We also sequenced 49 single clones isolated from 22 populations at the end of the 12 day experiment to determine mutation linkage.

### Evolution of CIP resistance

The rate and extent of evolved resistance depends on the strength of antibiotic selection (*Andersson and Hughes, 2014*; *Oz et al., 2014*), the distribution of fitness effects of mutations that increase resistance to the drug (*Maclean et al., 2010*), and the population size of replicating bacteria (*Salverda et al., 2017*; *Cooper, 2018*). The mode of bacterial growth can in principle alter each of these three variables and generate different dynamics and magnitudes of AMR. In the populations exposed to the increasing concentrations of CIP (the evolutionary rescue), the magnitude of evolved CIP resistance differed between planktonic and biofilm populations. Planktonic populations became approximately 160x more resistant on average than the ancestral clone while the biofilm populations

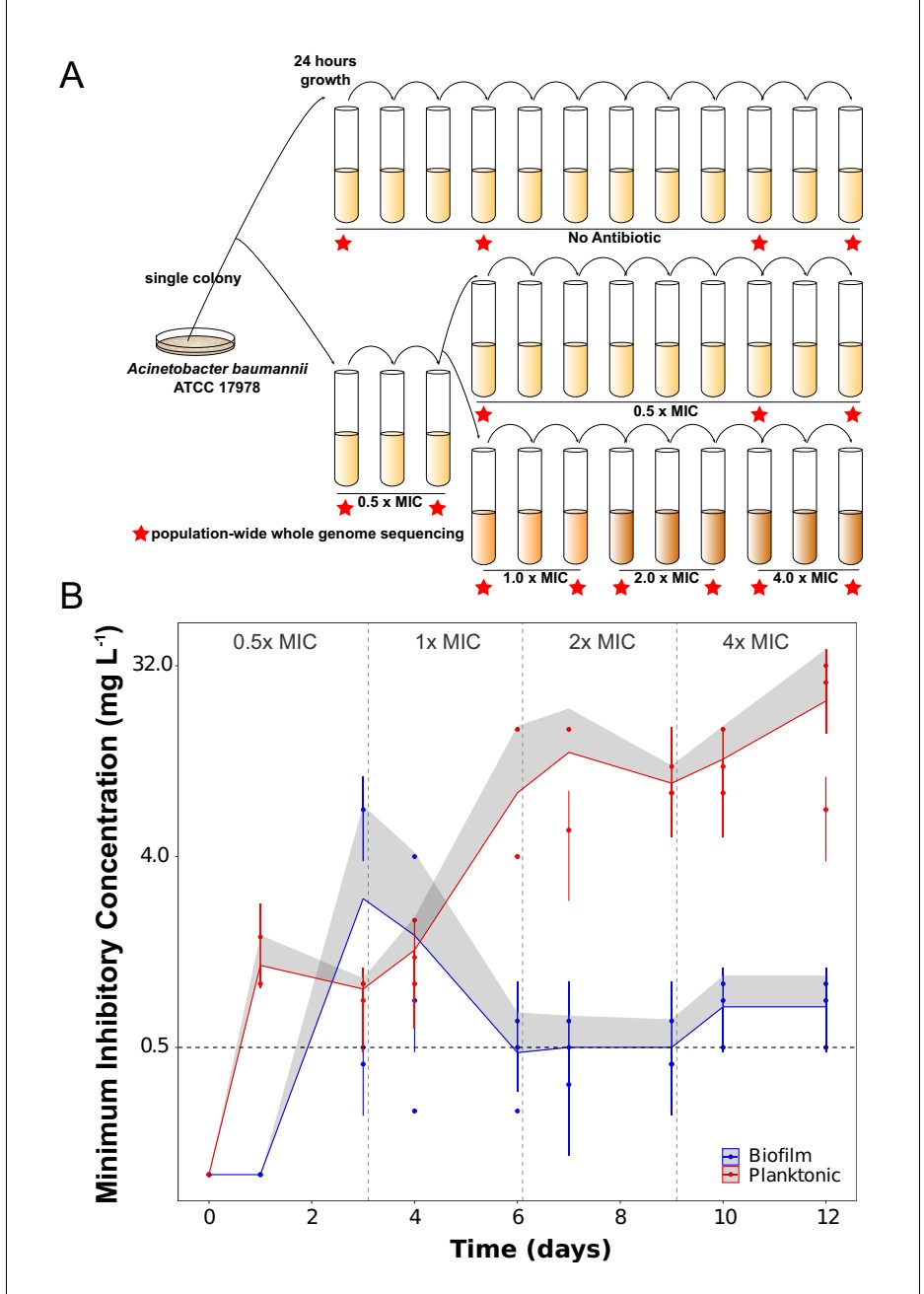

**Figure 1.** Experimental design (**A**) and dynamics of evolved resistance levels during the evolutionary rescue experiment (**B**). (**A**) A single clone of *A. baumannii* ATCC 17978 was propagated both in biofilm and planktonic conditions for 12 days under no antibiotics (top), subinhibitory concentrations of CIP (0.0625 mg/L = 0.5 x MIC) (middle) or in increasing concentrations of CIP (bottom). For the latter, termed evolutionary rescue, the concentration of CIP was doubled from 0.5 x MIC to 4.0 x MIC every 72 hr. As a control, five populations of *A. baumannii* ATCC 17978 were propagated in biofilm and five in planktonic conditions in the absence of antibiotics. We estimated the MICs to CIP and froze the populations for sequencing before and after doubling the antibiotic concentrations (red stars). (**B**) MICs (mg/L) of CIP were measured for replicate populations during the evolutionary rescue. The red and blue points represent the MICs of three populations propagated in planktonic or biofilm, respectively, with the 95% CI represented by the error bars. The red and blue lines represent the grand mean of the three planktonic and biofilm populations, respectively, with the upper 95% CI indicated by the gray shaded area. Horizontal dashed line indicates the highest CIP exposure during the experiment (4x MIC) and vertical lines indicate time when CIP concentration was doubled.

DOI: https://doi.org/10.7554/eLife.47612.003

became only 6x more resistant (*Figure 1B* and *Table 1*). Planktonic populations also evolved resistance much more rapidly, becoming 10x more resistant after only 24 hr of growth in sub-inhibitory CIP. This level of resistance would have been sufficient for surviving the remainder of the experiment, but MICs continued to increase at each sampling (*Figure 1B*). The evolution of resistance far beyond the selective requirement indicates that mutations conferring higher resistance also increased fitness in planktonic populations exposed to CIP.

In contrast, biofilm-evolved populations evolved under the evolutionary rescue regime acquired much lower levels of resistance (*ca.* 3– 7x the ancestral MIC) and primarily in a single step between days 3 and 4 (*Figure 1B*). In one notable exception, the MIC of biofilm population B2 increased ~50 x after 3 days of selection in subinhibitory concentrations of CIP (*Figure 1B*), but then the resistance of this population declined to only 6x higher than the ancestral strain. This dynamic suggested that a mutant conferring high-level resistance rose to intermediate frequency but was replaced by a more fit, yet less resistant, mutant (this possibility is evaluated below).

Lower levels of resistance were observed in populations selected at constant subinhibitory concentrations of CIP. Biofilm populations were 4x more resistant than the ancestor and planktonic

**Table 1.** Antibiotic susceptibility of the populations propagated in the absence, in subinhibitory concentrations or increasing concentrations of CIP at the end of the experiment (day 12).
MICs are expressed in mg/L and standard errors of the mean are indicated. Fold increases in MIC compared to the ancestral strain are also indicated.

| Treatment and Populations | MIC (mg/L) | Fold MIC increase |
| --- | --- | --- |
| No Antibiotic | | |
| Planktonic | | |
| P1 | 0.25 ± 0.00 | 2 |
| P2 | 0.25 ± 0 | 2 |
| P3 | 0.21 ± 0.03 | 1.68 |
| Biofilm | | |
| B1 | 0.25 ± 0 | 2 |
| B2 | 0.25 ± 0 | 2 |
| B3 | 0.25 ± 0 | 2 |
| Subinhibitory | | |
| Planktonic | | |
| P1 | 2.33 ± 0.72 | 18.64 |
| P2 | 4 ± 0 | 32 |
| P3 | 1 ± 0 | 8 |
| Biofilm | | |
| B1 | 0.41 ± 0.07 | 3.28 |
| B2 | 0.5 ± 0 | 4 |
| B3 | 0.5 ± 0 | 4 |
| Evolutionary rescue | | |
| Planktonic | | |
| P1 | 26.67 ± 5.34 | 213.36 |
| P2 | 32 ± 0 | 256 |
| P3 | 6.67 ± 1.34 | 53.36 |
| Biofilm | | |
| B1 | 1 ± 0 | 8 |
| B2 | 0.5 ± 0 | 4 |
| B3 | 0.83 ± 0.17 | 6.64 |

DOI: https://doi.org/10.7554/eLife.47612.004

populations were 20x more resistant (*Table 1*). We can infer that biofilm growth does not select for the high-level resistance seen in planktonic populations, instead favoring mutants with low levels of resistance and better adapted to life in a biofilm. It is important to note that these MIC measurements were made in planktonic conditions according to the clinical standards (*CLSI, 2019*) and that these values increased when measured in biofilm (*Table 2*). Our results correspond with studies of clinical isolates in which those producing more biofilm (and likely having adapted in biofilm conditions) were less resistant than non-biofilm-forming isolates (*Wang et al., 2018*). Nevertheless, antibiotic resistance levels are context-dependent (*Borriello et al., 2004*; *Hill et al., 2005*; *Kirby et al., 2012*), and because the biofilm environment at least partially protects cells from antibiotic exposure (*Table 2*), it can be argued that differences in MICs are due to the fact that evolved biofilm populations experienced lower CIP concentrations than planktonic populations. However, we selected CIP because it can penetrate the biofilm barrier (*Tseng et al., 2013*), and furthermore, cells growing in the bead model must disperse from one bead to colonize the next one in a less protected state. Overall, the fact that the planktonic populations exposed to subinhibitory concentrations of CIP increased their resistance level approximately 20-fold (*Table 1*) demonstrates that exposing bacteria to low levels of antibiotic risks selection for high levels of resistance that can make future treatment more difficult (*Wistrand-Yuen et al., 2018*).

## Evolutionary dynamics under CIP treatment

In large bacterial populations ($>10^5$ cells) growing under strong selection, adaptive mutations conferring beneficial traits such as antibiotic resistance will dominate population dynamics (*Barrick and Lenski, 2013*; *Cooper, 2018*). Therefore, if a single mutation renders the antibiotic ineffective and provides the highest fitness gain, it would be expected to outcompete all other less fit mutations. Further, the stronger the selection for resistance, the greater the probability of genetic parallelism among replicate populations (*Bolnick et al., 2018*). Under the population-genetic conditions of these experiments described above, a conservative estimate of $10^6$ mutations occur in the first growth cycle and at least $10^7$ mutations arise over the 12 days of selection, leading to a probability of 0.98 that every site in the 4Mbp *A. baumannii* genome experiences a mutation at least once over the course of the 12 day experiment (see *Supplementary file 1* for details of these calculations). Further, as stated above, fluoroquinolones like CIP or biofilm growth may increase the mutation rate so the probability that every site is mutated may be higher than estimated by this simplistic model (*Long et al., 2016*; *Geisinger et al., 2018*; *Pribis et al., 2019*). However, these studies do not indicate that fluoroquinolones like CIP alter the mutation spectra or particular mutation targets (*Long et al., 2016*), so a lifestyle-dependent difference in CIP exposure seems unlikely to alter the availability of resistance mutations under selection in these experiments. Rather, the dramatic

**Table 2.** Antibiotic susceptibility of one clone of each population propagated in increasing concentrations of CIP at the end of the experiment (day 12).

MICs were measured in biofilms and are expressed in mg/L and standard errors of the mean are indicated. Fold increase in MIC compared to the ancestral strain are also indicated.

| Treatment and Populations | MIC measured in biofilms (mg/L) | Fold MIC increase |
|---|---|---|
| Evolutionary rescue | | |
| Planktonic | | |
| P1 | >128 ± 0 | >1024 |
| P2 | >128 ± 0 | >1024 |
| P3 | >128 ± 0 | >1024 |
| Biofilm | | |
| B1 | 32 ± 0 | 256 |
| B2 | 32 ± 0 | 256 |
| B3 | 32 ± 0 | 256 |

DOI: https://doi.org/10.7554/eLife.47612.005

differences in the evolved resistance levels of planktonic and biofilm populations suggested distinct genetic causes of resistance produced by different selective forces that appear incongruent with mutation availability. We also predicted greater genetic diversity in the biofilm treatments, owing to spatial structure and/or niche differentiation (*Traverse et al., 2013*), than in the planktonic cultures, in which we expected selective sweeps (*Barrick et al., 2009*). A signature of spatial structure alone might be different mutations in the same gene with predicted similar function coexisting over time, which is a form of clonal interference (*de Visser and Rozen, 2006*). A signature of niche differentiation might be the coexistence of mutations in different genes with unique functions, which is a form of adaptive radiation (*Kassen, 2009*).

We conducted whole-population genomic sequencing of three replicates per treatment to identify all contending mutations above a detection threshold of 5% (see Materials and methods). The spectrum of mutations from CIP-treated populations are consistent with expectations from strong positive selection on altered or disrupted coding sequences (see *Table 3* for day 12 results and *Supplementary file 2* for dynamics across the experiment). High nonsynonymous to synonymous mutation ratios were observed in both lifestyles (8.5 in planktonic and 9.7 in biofilm). 43% of the total mutations in planktonic and 34% in biofilm were insertions or deletions, which is vastly enriched over typical mutation rates of ~10 SNPs/indel under neutral conditions (*Lynch et al., 2016*; *Dillon et al., 2017*). Roughly 30% of the mutations in CIP-treated populations of either lifestyle occurred in intergenic regions, which is statistically enriched over the approximately 13.5% of intergenic regions of the ancestral strain (X-squared = 8.2237, df = 2, p-value=0.01638). Of the intergenic mutations, 72% of the planktonic mutations and 18% of the biofilm mutations occurred in promoters, 5' untranslated regions, sRNAs or in putative terminators (*Kröger et al., 2018*) indicating that, as in *Pseudomonas,* intergenic mutations can be adaptive by regulating the transcription of different genes, while avoiding possible pleiotropic effects of mutations in the coding regions (*Khademi et al., 2019*).

As expected from theory, in CIP-selected planktonic populations where selection is most efficient, one or two mutations rapidly outcompeted others and fixed (*Figure 2*). Selection in biofilms, however, produced fewer selective sweeps and maintained more contending mutations, especially at lower antibiotic concentrations. In one population, multiple mutations in the same locus (*adeL*) rose to high frequency and persisted, which is consistent with the effect of population structure producing clonal interference. In the other two populations, mutations in different efflux pumps (*adeL, adeS, adeN*) contended during the experiment, which could be explained by population structure or ecological diversification, if these mutations produced different phenotypes. Overall, across all treatments and timepoints, biofilm-adapted populations were significantly more diverse than the planktonic-adapted populations (Shannon index; Kruskal Wallis, chi-squared = 7.723, p=0.005), particularly at subinhibitory concentrations of CIP (*Figure 2—figure supplement 1a*). Notably, increasing drug concentrations eliminated the differences in diversity between treatments (*Figure 2—figure supplement 1b*), but the greater diversity in biofilms treated with lower doses generated more diversity for selection to act upon in a changing environment. This higher standing

**Table 3.** Mutation spectrum of different selective environments.
Attributes of the contending mutations during the 12 days of the evolution experiment. [a]Results from the last day of the experimental evolution. [b]Accounting for all unique mutations detected after filtering (see Materials and methods). For mutation dynamics over time, see *Supplementary file 2*.

| | Increasing concentrations | | Subinhibitory concentrations | |
|---|---|---|---|---|
| | **Planktonic** | **Biofilm** | **Planktonic** | **Biofilm** |
| Total mutations | 28 | 38 | 6 | 16 |
| Nonsynonymous/Synonymous[a] | 8.5 | 9.67 | 2/0 | 6 |
| Intergenic | 8 | 11 | 0 | 4 |
| Nonsynonymous | 9 | 13 | 2 | 6 |
| Percent intergenic mutations[b] | 29 | 29 | 0 | 25 |

DOI: https://doi.org/10.7554/eLife.47612.006

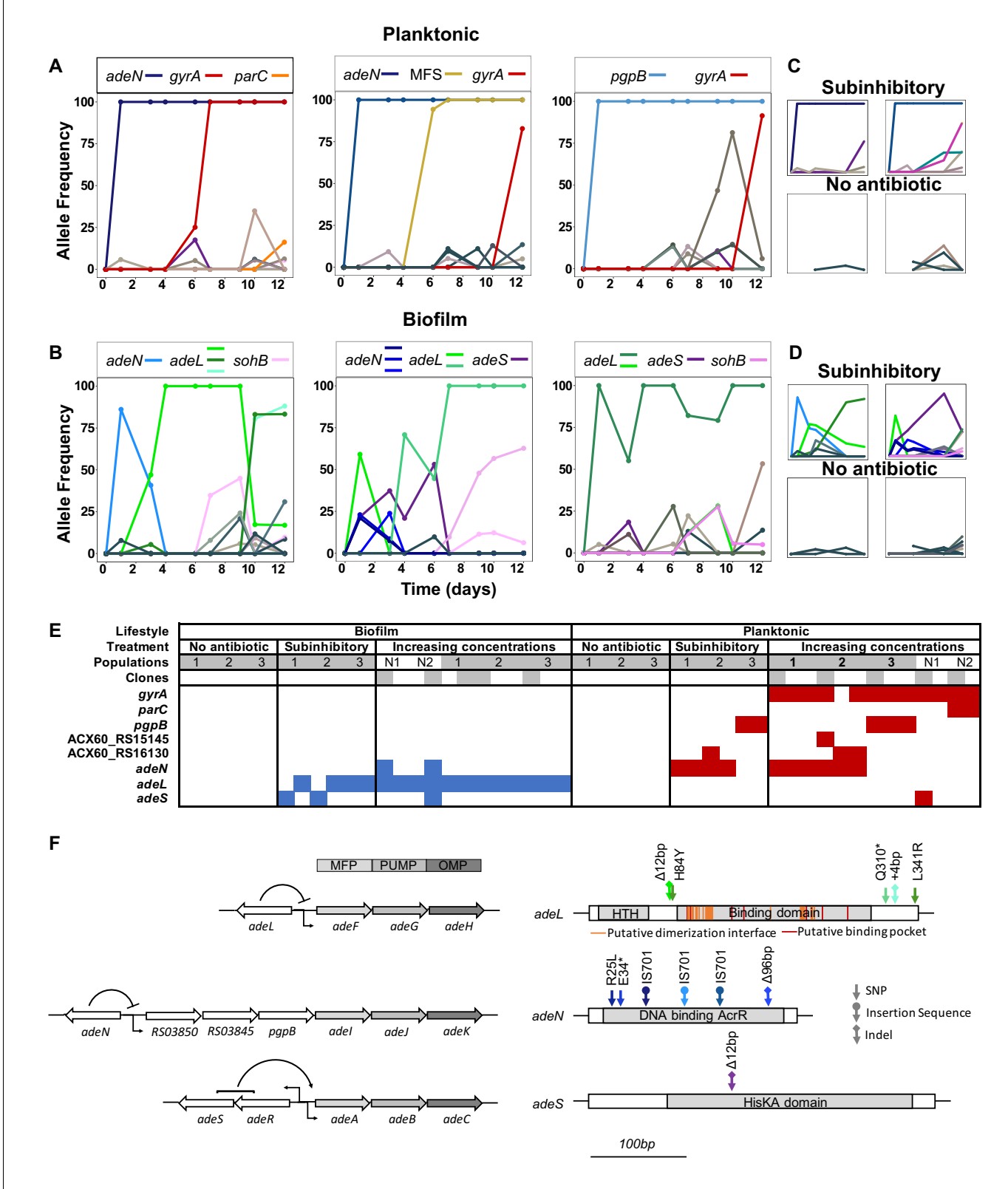

**Figure 2.** Lifestyle-dependent mutations and dynamics under increasing CIP selection. Mutation frequencies in planktonic (**A and C**) and biofilm populations (**B and D**) over time as determined by population-wide whole genome sequencing. **A)** and **B)** show the mutation frequencies obtained under increasing concentrations of CIP. From left to right: P1, P2 and P3 in A) and B1, B2 and B3 in B). **C)** and **D)** show the mutation frequencies obtained under the subinhibitory (top) and no antibiotic (bottom) treatments. Mutations in the same gene share a color. Blue: *adeN* or genes regulated

*Figure 2 continued on next page*

Figure 2 continued

by *adeN*; green: *adeL*; gold: MFS putative transporter ACX60_RS15145; purple: *adeS*; pink: *sohB*; red: *gyrA*; and orange: *parC*. Gray and brown colors indicate genes potentially unrelated to adaptation to CIP. (**F**) Mutated genes in the sequenced clones. Each column represents one clone. Gray shading of populations indicates whole population sequencing and N1 and N2 indicate populations where only clones were sequenced. Gray shaded clones were used for MIC and fitness estimations. Blue and red indicate SNPs in biofilm and planktonic growing populations respectively. For all SNPs identified in the 49 clones, see *Figure 2—figure supplement 2* and *Supplementary file 2*. (**G**) The genetic organization of the RND efflux pumps is shown on the left. MFP and OMP denote membrane fusion protein and outer membrane protein respectively. All mutations found in the RND regulators are shown on the right.

DOI: https://doi.org/10.7554/eLife.47612.007

The following source data and figure supplements are available for figure 2:

**Source data 1.** Mutation frequencies in evolved populations, labelled for example P4.60x = planktonic, replicate 4, day 6, 0x no antibiotic.

DOI: https://doi.org/10.7554/eLife.47612.011

**Figure supplement 1.** Genetic diversity of samples at subinhibitory concentrations of ciprofloxacin (**A**) or during the evolutionary rescue experiment with increasing concentrations of ciprofloxacin (**B**).

DOI: https://doi.org/10.7554/eLife.47612.008

**Figure supplement 2.** Mutated genes in the sequenced clones differ between treatments.

DOI: https://doi.org/10.7554/eLife.47612.009

**Figure supplement 3.** Biofilm production under subinhibitory concentrations of CIP.

DOI: https://doi.org/10.7554/eLife.47612.010

diversity is important when considering dosing and when antibiotic exposure may be low (*e.g.* in the external environment or when bound to tissues) (*Baquero et al., 1998*; *Khan et al., 2013*) because biofilms with more allelic diversity have a greater chance of survival to drug and immune attack (*Fux et al., 2005*).

In contrast with the data observed in the populations evolving under CIP pressure, drug-free control populations contained no mutations that achieved high frequency during the experiment (*Figure 2C and D*). These results suggest that the ancestral starting clone was already well adapted both to planktonic and biofilm lifestyles, likely because we had previously propagated the *A. baumannii* ATCC 17978 clone under identical drug-free conditions for 10 days leading to the fixation of mutations in three genes (*Supplementary file 2*). The absence of mutations specific to lifestyle in the absence of antibiotics and the acquisition of mutations specific to the growth mode under antibiotic pressure highlight different evolutionary responses to combined selective pressures that were not observed with each selective pressure alone (*Harrison et al., 2017*).

## Lifestyle determines the selected mechanisms of resistance

*A. baumannii* clinical samples acquire resistance to CIP by two principal mechanisms: modification of the direct antibiotic targets — gyrase A or B and topoisomerase IV — or by the overexpression of efflux pumps reducing the intracellular concentrations of the antibiotic (*Doi et al., 2015*). To directly associate genotypes with resistance phenotypes, we sequenced 49 clones isolated at the end of the experiment, the majority of which were selected to delineate genotypes in the evolutionary rescue populations (*Figure 2E* and *Figure 2—figure supplement 2*).

Both the genetic targets and mutational dynamics of selection in planktonic and biofilm environments differed. Mutations disrupting three negative regulators of efflux pumps evolved in parallel across populations exposed to CIP, but mutations in two of these (*adeL* and *adeS*) were nearly exclusive to biofilm clones (*Figure 2E*). The most common and highest frequency mutations observed in the biofilm populations were in the repressor gene *adeL* (*Figure 2E*, *Figure 2—figure supplement 2*, and *Table 4*), which regulates AdeFGH, one of three resistance-nodulation-division (RND) efflux pump systems in *A. baumannii* (*Coyne et al., 2010*; *Fernando et al., 2013*; *Pournaras et al., 2016*). The overexpression of the AdeFGH is predicted to enhance transport of acylated homoserine lactones, which can increase both biofilm and antibiotic resistance (*He et al., 2015*; *Alav et al., 2018*). In the planktonic lines, the predominant mutations were found in *adeN*, which is a negative regulator of AdeIJK and were mainly insertions of IS701 that disrupted the gene (*Li et al., 2016*). AdeIJK contributes to resistance to biocides, hospital disinfectants, and to both intrinsic and acquired antibiotic resistance in *A. baumannii* (*Damier-Piolle et al., 2008*; *Rosenfeld et al., 2012*) and may decrease

**Table 4.** Efflux pumps and their regulators in *A. baumannii* 17978 targeted under CIP pressure. Adapted from *Li et al. (2016)*. AG aminoglycosides, AZI azithromycin, BL β-lactams, CHL chloramphenicol, CL clindamycin, ERY erythromycin, FLO florfenicol, FUA fusidic acid, FQ fluoroquinolones, GEN gentamicin, MIN minocycline, NAL nalidixic acid, NOR norfloxacin, RIF rifampicin, SUL sulfonamides, TET tetracycline, TGC tigecycline, TMP trimethoprim. [a] References in *Li et al. (2016)*.

| Transporter Family | Regulator | Efflux pump | Substrates[a] |
|---|---|---|---|
| RND | AdeSR | AdeABC | AG, **BL**, CHL, ERY, **FQ**, NAL, TET, TGC |
| RND | AdeL | AdeFGH | CHL, ERY, **FQ**, NAL, SUL, TET, TGC, TMP |
| RND | AdeN | AdeIJK | AZI, **BL**, CHL, ERY, **FQ**, FUA, MIN, NAL, RIF, SA, SUL, TET, TMP |
| MATE | - | AbeM | **FQ**, GEN |

DOI: https://doi.org/10.7554/eLife.47612.012

biofilm formation, which could explain its prevalence in planktonic populations here (*Yoon et al., 2015*).

In biofilm lines, different contending *adeL* mutations were detected in each replicate after 24 hr then eventually fixed as CIP concentrations increased (green lines in *Figure 2B*), sometimes along with a secondary *adeL* mutation. This pattern suggests that altering efflux via *adeL* generates adaptations to the combination of CIP and biofilm which is supported by the increase in biofilm formation by the *adeL* mutants (*Figure 2—figure supplement 3*). Further, mutants with higher resistance than necessary appear to be maladaptive in the biofilm treatment. For example, *adeN* (found more often in planktonic culture) and *adeS* mutations found simultaneously on day three in population B2 (*Figure 2*) led to a spike in resistance at that timepoint (*Figure 1*), but these alleles were subsequently outcompeted by *adeL* mutants that were evidently more fit despite lower planktonic resistance.

In contrast to the biofilm populations, all planktonic populations with increasing concentrations of CIP eventually underwent selective sweeps of a single high frequency mutation in *gyrA* (S81L), the canonical ciprofloxacin-resistant mutation in DNA gyrase. These *gyrA* mutations evolved in genetic backgrounds containing either an *adeN* mutant or a *pgpB* mutant. *pgpB* is a gene that encodes a putative membrane associated lipid phosphatase and is co-regulated by *adeN* (*Hua et al., 2014*). Other mutations associated with high levels of resistance affected *parC,* encoding topoisomerase IV, and regulatory regions of two putative transporters, ACX60_RS15145 and ACX60_RS1613, the latter being co-transcribed with the multidrug efflux pump *abeM* (*Su et al., 2005*). Few other mutations exceeded the 10% of the total population filter in the planktonic lines. The repeated, rapid fixation of only *adeN* and *adeN*-regulated alleles in the planktonic CIP-exposed populations indicate that *adeN* conferred higher fitness than other CIP-resistant mutations at low drug concentrations or that these mutations were more accessible than others conferring resistance, though their mutation types do not support the latter interpretation (*Supplementary file 2*). Subsequently, at increased concentrations of CIP, on-target mutations in *gyrA* were favored in each line.

Together, our results demonstrate that bacterial lifestyle influences the evolutionary dynamics and targets of selection of AMR. Multiple selective pressures, particularly in the biofilm life cycle, may affect evolutionary dynamics and constrain the evolution of AMR if negative genetic correlations exist (*Harrison et al., 2017*). For instance, *adeN* mutations decrease biofilm formation and increase resistance by altering the *adeN*-controlled *adeIJK* efflux pump (*Yoon et al., 2015*), which could explain their prevalence in planktonic populations but not biofilm populations. In contrast, loss-of-function mutations in regulators of the *adeFGH* and *adeABC* RND efflux pumps were selected in CIP-treated biofilm populations and increased resistance ~4 fold, but these were not selected in planktonic populations perhaps because of this low resistance phenotype. Subsequent adaptation by planktonic populations exposed to CIP then selected mutations in the targets of the fluoroquinolone, *gyrA* and *parC*, leading to much higher levels of resistance.

## Evolutionary consequences of acquiring resistance

A longstanding hypothesis is that de novo acquired antibiotic resistance is associated with a fitness cost in the absence of antibiotics (reviewed in *Vogwill and MacLean, 2015*). The extent of this cost and the ability to compensate for it by secondary mutations (compensatory evolution) is a key attribute determining the spread and maintenance of the resistance mechanism (*Moore et al., 2000*; *Zhao and Drlica, 2002*; *Maclean et al., 2010*; *Vogwill and MacLean, 2015*). A negative correlation between CIP resistance and fitness of resistant genotypes in the absence of antibiotics has been previously described in *Escherichia coli,* suggesting a trade-off between these traits (*Marcusson et al., 2009*; *Huseby et al., 2017*; *Basra et al., 2018*).

To determine the relationship between resistance and fitness in the absence of antibiotics in our experiment, we chose 10 clones (five each from biofilm and planktonic populations, *Figure 2F* and *Figure 2—figure supplement 2*) with different genotypes and putative resistance mechanisms and measured their resistance and fitness phenotypes in both planktonic and biofilm conditions (*Figure 3*). As expected from the population mean values (*Figure 1B*), the biofilm clones much were less resistant in planktonic conditions than those evolved planktonically [MIC = 0.58 mg/L (SEM = 0.13) vs. MIC = 8.53 mg/L (SEM = 1.96), two-tailed t-test: p<0.05, t = 4.048, df = 80]. However, biofilm-evolved clones were more fit relative to the ancestral strain than the planktonic-evolved clones in the absence of antibiotic (two-tailed t-test: p=0.008, t = 2.984 df=18) (*Figure 3*). Importantly, these fitness measurements were made in both planktonic and biofilm conditions, demonstrating that even in the conditions they evolved in, and even following the preadaptation phase conducted in planktonic cultures, planktonic-selected clones were less fit as a result of fitness trade-offs of antibiotic resistance. However, one planktonic-evolved clone with mutations in both *gyrA* and *parC* exhibited no significant fitness cost and high levels of resistance. This suggests that, as in *Pseudomonas aeruginosa*, the *parC* mutation may compensate for the cost imposed by the *gyrA* mutation

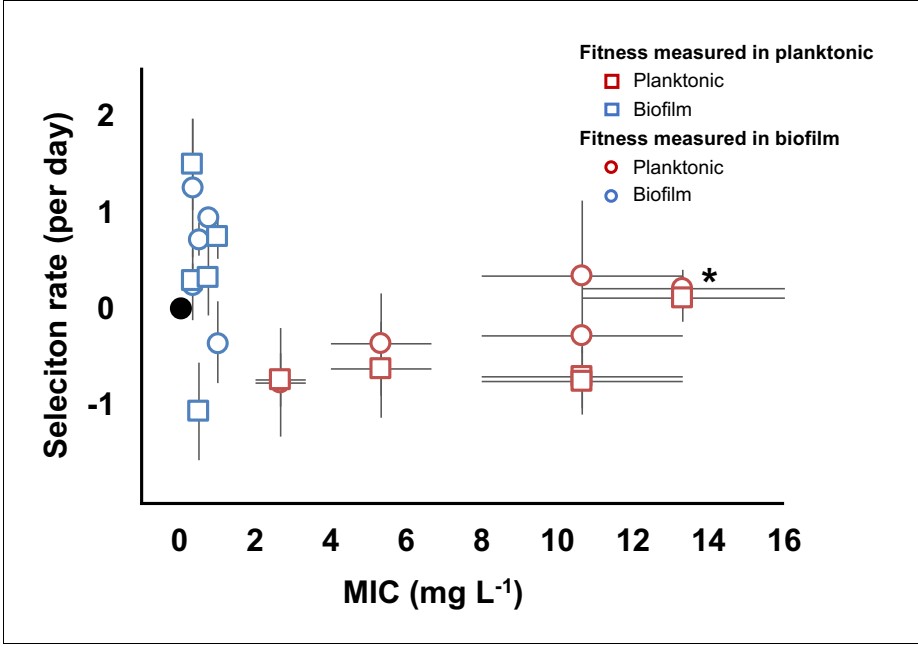

**Figure 3.** Evolved trade-off between resistance level and fitness. Relative fitness (average ± SEM) of 10 evolved clones from the evolutionary rescue experiment compared to the ancestor and their MICs (mg/L) to CIP. Fitness was measured in both planktonic (squares) and biofilm (circles) conditions. MICs were estimated in planktonic conditions. Black dot represents the ancestral clone. *Denotes the clone with *gyrA* and *parC* mutations.
DOI: https://doi.org/10.7554/eLife.47612.013

The following source data is available for figure 3:

**Source data 1.** Mean fitness and resistance values of selected clones.
DOI: https://doi.org/10.7554/eLife.47612.014

(*Kugelberg et al., 2005*), an example of sign epistasis (*Sackman and Rokyta, 2018*). Overall, mutants selected in biofilm-evolved populations were less resistant than mutants selected in planktonic populations (*Figure 1B*) but produced more biofilm (*Figure 2—figure supplement 3*) and paid little or no fitness cost in the absence of antibiotics (*Figure 3*). This cost-free resistance implies that these subpopulations could persist in the absence of drug, limiting the treatment options and demanding new approaches to treat high fitness, resistant pathogens (*Baym et al., 2016b*).

## Evolutionary interactions with other antibiotics

When a bacterium acquires resistance to one antibiotic, the mechanism of resistance can also confer resistance to other antibiotics (cross-resistance) or increase the susceptibility to other antibiotics (collateral sensitivity) (*Pál et al., 2015*). We tested the MIC of the evolved populations to 23 different antibiotics in planktonic conditions and reported quantitative changes in susceptibility by two-fold dilution, but not necessarily clinical breakpoints in resistance. Changes in susceptibilities were observed in 13 antibiotics that depended upon the growth mode of prior selection (*Figure 4*). For example, planktonic-evolved populations exhibited cross resistance to cefpodoxime and ceftazidime, but biofilm-evolved populations evolved collateral sensitivity to these cephalosporins. Cross-resistance was associated genetically with *adeN, adeS, gyrA* or *pgpB* mutations, and collateral sensitivity was associated with *adeL* mutations. Selection in these environments evidently favors the activation of different efflux pumps or modified targets that have different pleiotropic consequences for multidrug resistance (*Podnecky et al., 2018*).

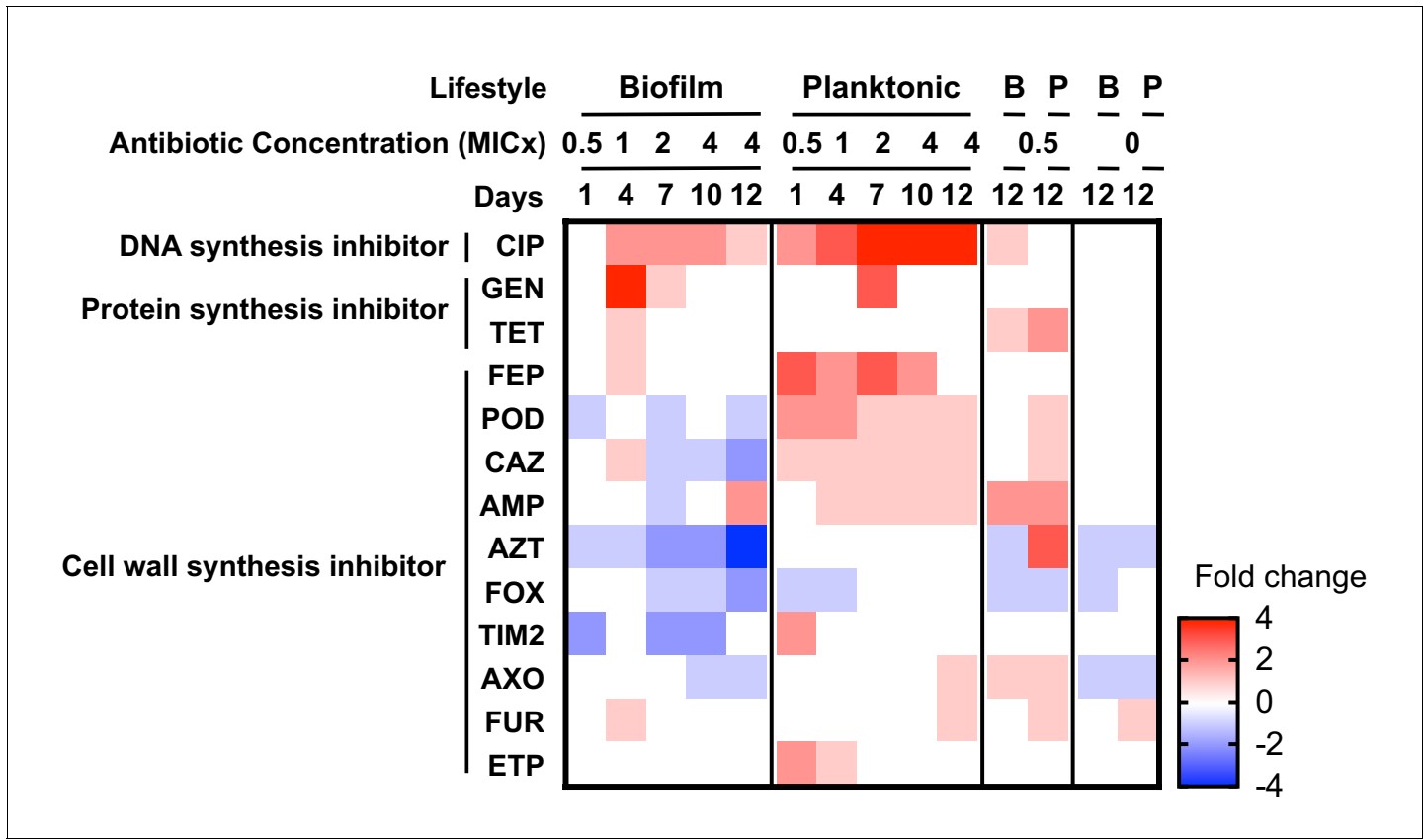

**Figure 4.** Collateral sensitivities and cross resistances to various antibiotics. Heat map showing the relative changes in antimicrobial susceptibility to 13 of the 23 antibiotics tested in the evolved populations (those not shown had no changes). Results shown are the median values of the fold change in the evolved populations compared to the ancestral strain. For subinhibitory and no-antibiotic treatments, only day 12 is shown. CIP, ciprofloxacin; GEN, gentamicin; TET, tetracycline; FEP, cefepime; POD, cefpodoxime; CAZ, ceftazidime; AMP, ampicillin; AZT, aztreonam; FOX, cefoxitin; TIM2, ticarcillin/clavulanic acid constant; AXO, ceftriaxone; FUR cefuroxime; ETP, ertapenem.
DOI: https://doi.org/10.7554/eLife.47612.015

The mechanisms leading to collateral sensitivity are still poorly understood but they depend on the genetic background of the strain, the nature of the resistance mechanisms (*Barbosa et al., 2017*; *Yen and Papin, 2017*), and the specific physiological context of the cells (*Leus et al., 2018*). In *A. baumannii*, each RND efflux pump is suggested to be specific for certain classes of antibiotics (*Table 4*) (*Coyne et al., 2011*; *Li et al., 2016*; *Leus et al., 2018*). Similar to our results (*Figure 4*), Yoon and collaborators demonstrated that efflux pumps AdeABC and AdeIJK, regulated by *adeS* and *adeN* respectively, increased the resistance level to some β-lactams when overexpressed (*Yoon et al., 2015*). However, production of AdeFGH, the efflux pump regulated by *adeL*, decreased resistance to some β-lactams and other families of antibiotics or detergents by an unknown mechanism (*Yoon et al., 2015*; *Leus et al., 2018*). Increased sensitivity to β-lactams with efflux overexpression has also been reported in *P. aeruginosa* (*Azimi and Rastegar Lari, 2017*), which demonstrates the urgency of understanding the physiological basis of collateral sensitivity to control AMR evolution. Exploiting collateral sensitivity has been proposed to counteract the evolution of resistant populations both in bacteria (*Imamovic and Sommer, 2013*; *Kim et al., 2014*; *Nichol et al., 2019*) and in cancer (*Dhawan et al., 2017*). Remarkably, our results show that biofilm growth, commonly thought to broaden resistance, may actually generate collateral sensitivity during treatment with CIP and potentially other fluoroquinolones.

## Conclusions

We used experimental evolution of the opportunistic pathogen *A. baumannii* in both well-mixed and biofilm conditions to examine how lifestyle influences the dynamics, diversity, identity of genetic mechanisms, and direct and pleiotropic effects of resistance to a common antibiotic (*Figure 5*). Experimental evolution is a powerful method of screening naturally arising genetic variation for mutants that are the best fit in a defined condition (*Elena and Lenski, 2003*; *Cooper, 2018*; *Van den Bergh et al., 2018*). When population sizes are large and reproductive rates are rapid, as they were here, the probability that all possible single-step mutations that can increase both resistance and fitness occurred in each population is very likely. The few mutations selected here as well

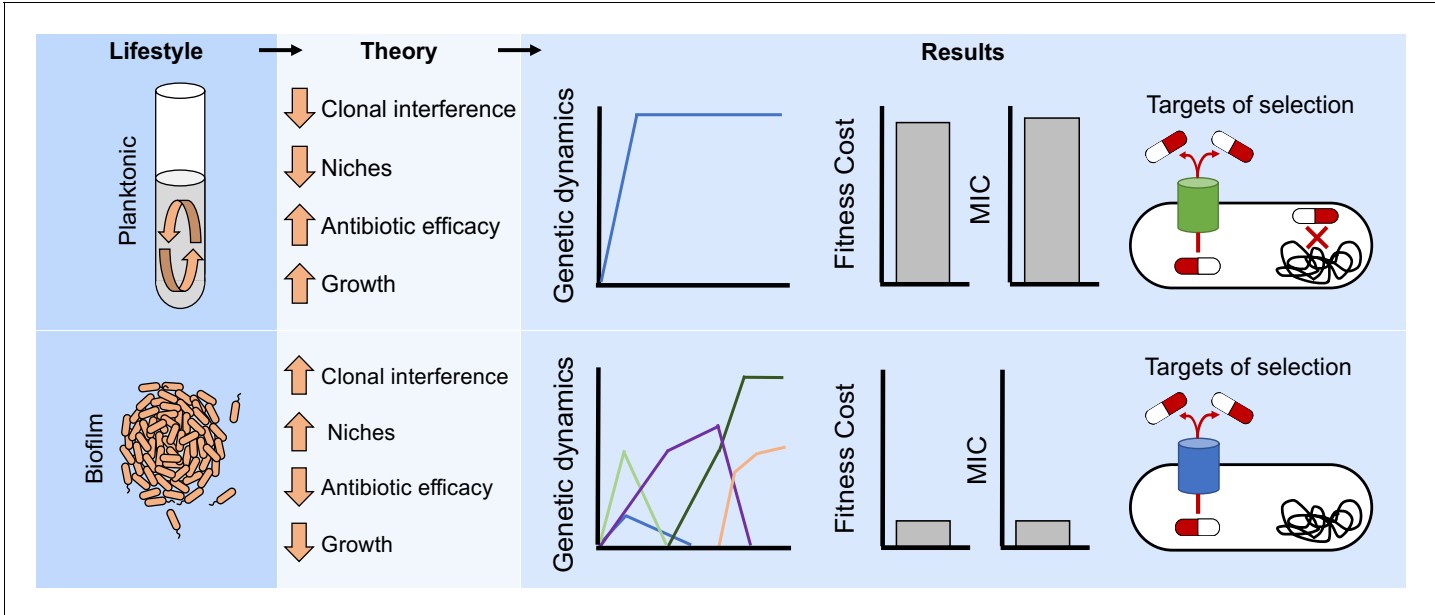

**Figure 5.** Lifestyle determines the evolution of antibiotic resistance. The environmental structure of biofilms can increase clonal interference and generate distinct ecological conditions which can favor different niches. Those factors will generate more complex evolutionary dynamics in biofilms compared to well-mixed populations (*Figure 2A–D* and *Figure 2—figure supplement 1*). In addition, the efficacy of the antibiotic can be reduced by the protection conferred by the biofilm and the relatively slow growth of the cells. Therefore, different evolutionary dynamics (*Figure 3*) and targets of selection conferring lower antibiotic resistance (*Figures 1B* and *2E* and *Figure 2—figure supplement 2*) will be selected for in structured populations compared to well-mixed ones.

DOI: https://doi.org/10.7554/eLife.47612.016

as their repeated order with increasing CIP concentrations may indicate that these are the most fit mutations in this *A. baumannii* strain and set of environmental conditions. The prevalence of some of these mutations in clinical samples suggests that they too may have been exposed to selection in similar conditions. For instance, S81L in *gyrA* and S80L *in parC* have been reported worldwide as the primary mechanism conferring high levels of resistance to fluroquinolones in clinical isolates (*Adams-Haduch et al., 2008*; *Warner et al., 2016*; *Dahdouh et al., 2017*), and mutations in RND efflux pumps have been associated with multidrug resistant phenotypes in clinical samples isolated world-wide (*Damier-Piolle et al., 2008*; *Coyne et al., 2010*; *Rosenfeld et al., 2012*; *Fernando et al., 2013*; *Pournaras et al., 2016*; *Leus et al., 2018*). Likewise, the absence of other mutations reported in shotgun mutant screens of resistance in *A. baumannii* (*Geisinger et al., 2018*) means that these mutants produced less resistance, lower fitness, or both.

Evolution experiments hold promise for ultimately forecasting mutations selected by different antimicrobials and anticipating treatment outcomes, including the diversification of the pathogen population and the likelihood of collateral sensitivity or cross-resistance (*Brockhurst et al., 2019*). Furthermore, knowledge of the prevailing lifestyle of the pathogen population may be critically important for treatment design. As predicted by our experiment, biofilm-forming clinical isolates were more susceptible to CIP and other antibiotics than non-biofilm forming clinical isolates even when the resistance levels were measured in biofilms (*Rodríguez-Baño et al., 2008*; *Qi et al., 2016*; *Wang et al., 2018*). Most infections are likely caused by surface-attached populations (*Wolcott et al., 2010*; *Wolcott, 2017*), and some treatments include cycling antibiotics that promote biofilm as a primary response. For example, tobramycin is used for treating *P. aeruginosa* in cystic fibrosis patients (*Hamed and Debonnett, 2017*) and promotes biofilm formation (*Hoffman et al., 2005*; *Linares et al., 2006*), wherein the evolution of antibiotic resistance without a detectable fitness cost may arise during treatment. As in our experiment, the overexpression of RND efflux pumps in both *P. aeruginosa* and *Neisseria gonorrhoeae* may produce little or no fitness cost (*Warner et al., 2007*; *Olivares Pacheco et al., 2017*). But the more diverse biofilm-adapted lineages in our experiments revealed a striking vulnerability to cephalosporins, which could provide a new strategy for treatment. Broader still, conventional wisdom has long held that the relationship between resistance and fitness is antagonistic, and that the efficacy of many antimicrobials is aided by a severe fitness cost of resistance (*Vogwill and MacLean, 2015*; *Baym et al., 2016b*; *Hughes and Andersson, 2017*). This study demonstrates that the form of the relationship between fitness and resistance can be altered by the mode of growth, whereby biofilms can align resistance and fitness traits. Continued efforts to determine how the fitness landscape of various resistance pathways depends on the environment and its structure, including growth mode, could produce a valuable forecasting tool to stem the rising AMR tide.

## Materials and methods

### Experimental evolution

Before the start of the antibiotic evolution experiment, we propagated well mixed tubes founded by one clone of the susceptible *A. baumannii* strain ATCC 17978-mff (*Piechaud and Second, 1951*; *Baumann et al., 1968*) in a modified M9 medium (referred to as M9$^+$) containing 0.1 mM CaCl$_2$, 1 mM MgSO$_4$, 42.2 mM Na$_2$HPO$_4$, 22 mM KH2PO$_4$, 21.7 mM NaCl, 18.7 mM NH$_4$Cl and 11.1 mM glucose and supplemented with 20 mL/L MEM essential amino acids (Gibco 11130051), 10 mL/L MEM nonessential amino acids (Gibco 11140050), and 1 mL each of trace mineral solutions A, B, and C (Corning 25021–3 Cl). This preadaptation phase was conducted in the absence of antibiotics for 10 days (*ca.* 66 generations) with a dilution factor of 100 per day.

After the ten days of preadaptation to M9$^+$ medium, we selected a single clone and propagated for 24 hr in M9$^+$ in the absence of antibiotic. We then subcultured this population into twenty replicate populations. Ten of the populations (5 planktonic and five biofilm) were propagated every 24 hr in constant subinhibitory concentrations of CIP, 0.0625 mg/L, which corresponds to 0.5x the minimum inhibitory concentration (MIC). After 72 hr under subinhibitory concentrations of CIP, the populations were exposed to two different antibiotic regimes for nine more days, either constant subinhibitory concentrations of CIP or increasing concentrations of CIP (called the evolutionary

rescue). For the latter, we doubled the CIP concentrations every 72 hr until 4x MIC. As a control, the 10 remaining populations were propagated in the absence of CIP (*Figure 1*).

We propagated the populations into fresh media every 24 hr as described by *Turner et al. (2018)*. For planktonic populations, we transferred a 1:100 (50 μl into 5 mL of M9$^+$) dilution, which corresponded to 6.64 generations per day. For biofilm populations, we transferred a polystyrene bead (Polysciences, Inc, Warrington, PA) to fresh media containing three sterile beads. We rinsed each bead in PBS before the transfer, therefore reducing the transfer of planktonic cells. Each day we alternated between black and white marked beads, ensuring that the bacteria were growing on the bead for 24 hr, which corresponds to approximately 6 to 7.5 generations/day (*Traverse et al., 2013*; *Turner et al., 2018*). For the experiment with increasing concentrations of antibiotics, we froze a sample of each bacterial population on days 1, 3, 4, 6, 7, 9, 10 and 12. In the experiment with constant exposure to subinhibitory concentrations of antibiotics, we froze the populations on days 1, 3, 4, 9, and 12. We froze the control populations at days 1, 4, 9, and 12. For planktonic populations, we froze 1 mL of culture with 9% of DMSO. For freezing the biofilm populations, we sonicated the beads in 1 mL of PBS with a probe sonicator and subsequently froze with 9% DMSO.

## Phenotypic characterization: antimicrobial susceptibility and biofilm formation

We determined the MIC of CIP by broth microdilution in M9$^+$ according to the Clinical and Laboratory Standards Institute guidelines (*CLSI, 2019*), in which each bacterial sample was tested to 2-fold-increasing concentration of CIP from 0.0625 to 64 mg/L. To obtain a general picture of the resistance profiles we determined the MIC to 23 antibiotics (amikacin, ampicillin, ampicillin/sulbactam, aztreonam, cefazolin, cefepime, cephalothin, meropenem, ertapenem, cefuroxime, gentamicin, CIP, piperacillin/tazobactam, cefoxitin, trimethoprim/sulfamethoxazole, cefpodoxime, ceftazidime, tobramycin, tigecycline, ticarcillin/clavulanic acid, ceftriaxone and tetracycline) by broth microdilution in commercial microtiter plates following the instructions provided by the manufacturers (Sensititre GN3F, Trek Diagnostics Inc, Westlake, OH). We tested the MIC at days 1, 3, 4, 6, 7, 9, 10 and 12 for the populations propagated under increasing concentrations of antibiotic, and at days 1 and 12 for the subinhibitory and non-antibiotic treatments. For the CIP-MICs, we used *Pseudomonas aeruginosa* PAO1 in Mueller Hinton broth as a control. No differences in the MICs were found between Mueller Hinton and M9$^+$ or if measuring the MIC in 96 well-plate or in 5 mL tubes, which are the experimental conditions. Each MIC was performed in triplicate. The CIP was provided by Alfa Aesar (Alfa Aesar, Wardhill, MA). We also determined the MIC of CIP in biofilm conditions adapting the method described by Diez-Aguilar to the bead model (*Díez-Aguilar et al., 2018*). We resuspended each clone into fresh M9$^+$ containing sterile beads (as in the experimental evolution conditions, each tube used contained three sterile beads and 5 mL of M9$^+$). After 24 hr growing at 37°, each bead was propagated into new fresh M9$^+$ containing different CIP concentrations (from 4 to 128 mg/L in 2-fold-increasing manner). After 24 growing at 37°, we rinsed each bead in PBS and sonicate them individually as explained before. 10 μl of the sonicated liquid were transferred to 100 μL of M9$^+$. The MIC was calculated after measuring the optical density at 650 nm before and after 24 hr incubation. The inhibition of growth was defined as the lowest antibiotic that resulted in an OD difference at or below 0.05 after 6 hr of incubation.

We estimated the biofilm formation of the selected clones using a modification of the previously described protocol (*O'Toole and Kolter, 1998*). We resurrected each clone in 5 mL of M9$^+$ containing 0.5 mg/L of CIP and grew them for 24 hr. For each strain, we transferred 50 μl into 15 mL of M9$^+$. We tested 200 μl of the previous dilution of each clone to four different subinhibitory CIP concentrations (0 mg/L, 0.01 mg/L, 0.03 mg/L and 0.0625 mg/L). After 24 hr of growing at 37°C, we measured population sizes by optical density (OD) at 590 nm (OD$_{Populations}$). Then, we added 250 μl of 0.1% crystal violet and incubated at room temperature for 15 min. After washing the wells and drying for 24 hr, we added 250 μl 95% EtOH solution (95% EtOH, 4.95% dH2O, 0.05% Triton X-100) to each well and incubated for 15 min and biofilm formation was measured by the OD at 590 nm (OD$_{Biofilm}$). Biofilm formation was corrected by population sizes (OD$_{Biofilm}$/OD$_{Population}$). Results are the average of three experiments (*Figure 2—figure supplement 3*).

## Fitness measurement

We selected 5 biofilm and five planktonic clones from the end of the evolutionary rescue experiment with known genotype (*Figure 2—figure supplement 2*) and measured their fitness by directly competing the ancestral strain and the evolved clone variants both in planktonic and in biofilm conditions in the absence of antibiotic (*Figure 3*) (*Turner et al., 2018*). We revived each clone from a freezer stock in M9$^+$ for 24 hr. We maintained the same evolutionary conditions to revive the clones, adding three beads and/or CIP to the broth when required. After 24 hr, we added equal volume of the clones and the ancestors in M9$^+$ in the absence of antibiotics. For planktonic populations, we mixed 25 µl of each competitor in 5 mL of M9$^+$. For biofilm competitions, we sonicated one bead per competitor in 1 mL of PBS and mixed in 5 mL of M9$^+$ containing three beads. The mix was cultured at 37°C for 24 hr. We plated at time zero and after 24 hr. For each competition, we plated aliquots onto nonselective tryptic soy agar and tryptic soy agar containing CIP. Selection rate (*r*) was calculated as the difference of the Malthusian parameters for the two competitors: $r = (\ln(\text{CIP resistant}_{d=1}/\text{CIP resistant}_{d=0}))/(\ln(\text{CIP susceptible}_{d=1}/\text{CIP susceptible}_{d=0}))/\text{day}$ (*Lenski, 1991*). Susceptible populations were calculated as the difference between the total population (number of colonies/mL growing on the nonselective plates) and the resistant fraction (number of colonies/mL growing on the plates containing CIP). As a control for calculating the correct ratio of susceptible vs. resistant populations, we replica-plated 50 to 100 colonies from the nonselective plates onto plates containing CIP as previously described (*Santos-Lopez et al., 2017*). Results are the average of three to five independent experiments.

## Genome sequencing

We sequenced whole populations of three evolving replicates per treatment. We sequenced the populations at days 1, 3, 4, 6, 7, 9, 10, and 12 of the populations under increasing concentrations of CIP (populations P1, P2, P3 and B1, B2, B3 for planktonic and biofilm populations) and at days 1, 4, 9, and 12 of the populations under subinhibitory concentration and no antibiotic treatments. In addition, we selected 49 clones for sequencing at the end of the experiment (*Figure 2F*). 12 of the clones were recovered from the populations propagated in the absence of the antibiotic, 12 clones from the subinhibitory concentrations of CIP treatment and 25 were isolated from the increasing concentrations of antibiotic. We revived each population or clone from a freezer stock in the growth conditions under which they were isolated (*i.e.* the same CIP concentration which they were exposed to during the experiment) and grew for 24 hr. DNA was extracted using the Qiagen DNAeasy Blood and Tissue kit (Qiagen, Hiden, Germany). The sequencing library was prepared as described by Turner and colleagues (*Turner et al., 2018*) according to the protocol of *Baym et al. (2015)*, using the Illumina Nextera kit (Illumina Inc, San Diego, CA) and sequenced using an Illumina NextSeq500 at the Microbial Genome Sequencing center (http://migs.pitt.edu).

## Data processing

All sequences were first quality filtered and trimmed with the Trimmomatic software v0.36 (*Bolger et al., 2014*) using the criteria: LEADING:20 TRAILING:20 SLIDINGWINDOW:4:20 MINLEN:70. Variants were called with the breseq software v0.31.0 (*Deatherage and Barrick, 2014*) using the default parameters and the -p flag when required for identifying polymorphisms in populations. This option calls a mutation if it is observed in two reads from each strand and reaches 5% in the population. The average depth of sequencing for populations was 219 ± 51 x and average genome coverage was 98.7 ± 0.128%. The reference genome used for variant calling was downloaded from the NCBI RefSeq database using the 17-Mar-2017 version of *A. baumannii* ATCC 17978-mff complete genome (GCF_001077675.1). In addition to the chromosome NZ_CP012004 and plasmid NZ_CP012005 sequences, we added two additional plasmid sequences to the reference genome that are known to be present in our working strain of *A. baumannii* ATCC 17978-mff: NC009083, NC_009084. Mutations were then manually curated and filtered to remove false positives. Mutations were filtered if the gene was found to contain a mutation when the ancestor sequence was compared to the reference genome or if a mutation never reached a cumulative frequency of 10% across all replicate populations. Diversity measurements were made in R using the Shannon index considering the presence, absence, and frequency of alleles. Significant differences between biofilm and planktonic populations were determined by the Kruskal Wallis test. Filtering,

mutational dynamics, and plotting were done in R v3.4.4 (www.r-project.org) with the packages ggplot2 v2.2.1 (*Wickham, 2016*), dplyr v0.7.4 (*Wickham et al., 2018*), vegan v2.5–1 (*Oksanen et al., 2018*), and reshape2 (*Wickham, 2007*).

## Data availability

R code for filtering and data processing can be found here: https://github.com/sirmicrobe/U01_allele_freq_code (*Santos-Lopez, 2019*; copy archived at https://github.com/elifesciences-publications/U01_allele_freq_code). All sequences were deposited into NCBI under the Biosample accession numbers SAMN09783599-SAMN09783682.

## Acknowledgements

We thank Caroline B Turner for helpful discussions and proofreading of the paper, Allison L Welp for laboratory assistance and Christopher Deitrick for depositing the sequences in the NCBI database. This research was supported by NIH U01AI124302-01.

## Additional information

### Funding

| Funder | Grant reference number | Author |
|---|---|---|
| National Institutes of Health | U01AI124302-01 | Vaughn S Cooper |

The funders had no role in study design, data collection and interpretation, or the decision to submit the work for publication.

### Author contributions

Alfonso Santos-Lopez, Conceptualization, Formal analysis, Validation, Investigation, Methodology, Writing—original draft, Writing—review and editing; Christopher W Marshall, Conceptualization, Data curation, Software, Formal analysis, Validation, Investigation, Methodology, Writing—original draft, Writing—review and editing; Michelle R Scribner, Investigation, Methodology; Daniel J Snyder, Resources, Methodology; Vaughn S Cooper, Conceptualization, Supervision, Funding acquisition, Investigation, Methodology, Project administration, Writing—review and editing

### Author ORCIDs

Alfonso Santos-Lopez (ID) https://orcid.org/0000-0002-9163-9947
Vaughn S Cooper (ID) https://orcid.org/0000-0001-7726-0765

### Decision letter and Author response

Decision letter https://doi.org/10.7554/eLife.47612.023
Author response https://doi.org/10.7554/eLife.47612.024

## Additional files

### Supplementary files

• Supplementary file 1. Estimated mutation probabilities during experimental evolution.
DOI: https://doi.org/10.7554/eLife.47612.017

• Supplementary file 2. Complete list of mutated genes in the pre-adapted ancestor, in the sequenced clones, and those detected within each population by day.
DOI: https://doi.org/10.7554/eLife.47612.018

• Transparent reporting form
DOI: https://doi.org/10.7554/eLife.47612.019

## Data availability

Sequencing data were deposited to NCBI as Bioproject 485123. R code for filtering and data processing can be found here: https://github.com/sirmicrobe/U01_allele_freq_code (copy archived at https://github.com/elifesciences-publications/U01_allele_freq_code).

The following dataset was generated:

| Author(s) | Year | Dataset title | Dataset URL | Database and Identifier |
|---|---|---|---|---|
| Santos-Lopez A, Marshall CW, Scribner MR, Snyder D, Cooper VS | 2019 | Evolutionary pathways to antibiotic resistance are dependent upon environmental structure and bacterial lifestyle. | https://www.ncbi.nlm.nih.gov/bioproject/485123 | NCBI BioProject, 485123 |

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
