## [Decision Letter]

Thank you for submitting your article "Evolutionary pathways to antibiotic resistance are dependent upon environmental structure and bacterial lifestyle" for consideration by *eLife*. Your article has been reviewed by Karla Kirkegaard as the Senior Editor and Reviewing Editor, and three reviewers. The following individual involved in review of your submission has agreed to reveal their identity: Ellie Margolis (Reviewer #1). The reviewers have discussed the reviews with one another and the Reviewing Editor has drafted this decision to help you prepare a revised submission.

Little is known about the evolutionary dynamics of antimicrobial resistance in biofilms – a prevalent mode of bacterial growth during infections-as most experimental work to date has focused on planktonic cultures. Santos-Lopez, Marshall, et al. begin to address this gap here by comparing experimental evolution to ciprofloxicin (CIP) resistance in Acinetobacter baumanii biofilm and planktonic cultures and showing clear differences between the adaptive pathways taken in different growth conditions. Intriguingly, the authors show that the primary mechanisms of CIP resistance differ between biofilm and planktonic evolution experiments and that the different resistance mechanisms are associated with different fitness levels and different susceptibility profiles to other antimicrobials. The methods and analysis are technically sound and the results are interesting.

Essential revisions:

All reviewers thought that the work was very interesting but that some of the conclusions were overstated based on the data presented. Therefore, either many of the conclusions should be tempered substantially or additional experiments should be done to allow for more definitive and mechanistic conclusions. The major concern seemed to be that, which each conclusion, all assumptions and caveats should be mentioned. For example:

1) The authors conclude that fitness rather than antibiotic resistance was selected for in the biofilm (Figure 3). Another interpretation is that the experimental setup could have imposed a conflicting selective pressure (complicated lifestyle of attachment, maturation and dispersion) from the antibiotic selection. Conflicting selection can have large impacts on diversity and evolutionary trajectory (especially when pleiotropic effects of mutations are occurring)-see Harrison et al., 2017 for example. This could also be a potential explanation for the poor fitness in biofilm of adeN and adeS mutations (subsection “Lifestyle determines the selected mechanisms 230 of resistance”). Other alternatives not explicitly stated would be slight decrease in antibiotic penetration/functionality (could have large difference in selective pressure) or fitness of planktonic was already at a local optima (constrained to evolve farther in experimental setup). Mention of these kinds of concerns would be appropriate.

2) In the clinical relevance section, the choice to show Figure 4 as a heat map for fold increase of susceptibilities assumes that antibiotic concentrations (MICs) are meaningful on a linear scale. This assumption should be stated.

3) Many of the conclusions of this study rely on the calculation (Supplementary file 3) that all possible mutations were sampled at the same frequency by each population during the course of the evolution experiments and therefore, any differences in evolutionary outcome is due to selection (Results and Discussion section and elsewhere). However, the calculations underlying this assumption rely on two key assumptions that where not experimentally tested here and have been refuted in other scenarios: that mutations occur at a fixed rate and that mutations occur with a uniform distribution at all genomic sites.

4) One reviewer stated this another way, with references, stating that rates are known to vary with environmental/lifestyle differences in bacteria and other organisms (reviewed Fitzgerald, 2019). Specifically, mutation rates are increased dramatically by CIP and other quinolone antibiotics (Cirz 2005; Kohanski, 2010; Pribis, 2019) and may be different between planktonic and biofilm cultures (Boles, 2004, 2008). Additionally, the intrinsic physical protection of biofilm growth may mean that the mutation rate induction by CIP is different between growth conditions. Without direct measurement of mutation rates in the relevant growth conditions (CIP doses, biofilm, planktonic), it is impossible to negate the possibility that mutation rates differ greatly between treatments and that these differences could contribute substantially to the evolved outcomes.

Thus, either mutation rates in unselected genes should be measured experimentally or the assumptions mentioned each time they are invoked and the conclusions toned down. As an example, the Results and Discussion section, the observation of rapid fixation of the adeN alleles and not the gyrA and parC alleles does not necessarily "indicate" that they confer higher resistance. Resistance-conferring adeN and related mutations could simply occur more frequently (higher local mutation rate, a larger target size, and/or have fewer selective constraints for other gene functions) compared to the topoisomerase mutants that emerge at later timepoints/higher [CIP]. The interpretation/conclusion needs to be softened or additional experiments (competition between adeN and gyrA/parC at low [CIP]) are needed to support it.

5) The lack of preadaptation to biofilm growth could be a major confounding factor in the comparison of adaptive paths taken in the two growth conditions. The data presented in Figure 2—figure supplement 3 suggest that the biofilm cultures were under additional selective pressure to adapt to biofilm growth and, therefore, a mutation that improved both biofilm formation and CIP resistance (such as adeL) would be highly preferred to one that only improved CIP resistance.

It is possible that an alternative experimental design involving preadaptation to the experimental growth condition (planktonic or biofilm) prior to CIP exposure would yield quite different adaptive trajectories, helping to separate the selective pressure of the antibiotic from that of the growth condition. In the absence of such additional experimentation, this possibility should be explicitly mentioned.

6) In Figure 4, please add some details about the qualities of these antibiotics. On the Y axis, do they cluster with targets at all? More explanation of the targets of each antibiotics is needed, because not all readers will know the targets.

---

## [Author Response]

All reviewers thought that the work was very interesting but that some of the conclusions were overstated based on the data presented. Therefore, either many of the conclusions should be tempered substantially or additional experiments should be done to allow for more definitive and mechanistic conclusions. The major concern seemed to be that, which each conclusion, all assumptions and caveats should be mentioned. For example:1) The authors conclude that fitness rather than antibiotic resistance was selected for in the biofilm (Figure 3). Another interpretation is that the experimental setup could have imposed a conflicting selective pressure (complicated lifestyle of attachment, maturation and dispersion) from the antibiotic selection. Conflicting selection can have large impacts on diversity and evolutionary trajectory (especially when pleiotropic effects of mutations are occurring)-see Harrison et al., 2017 for example. This could also be a potential explanation for the poor fitness in biofilm of adeN and adeS mutations (subsection “Lifestyle determines the selected mechanisms 230 of resistance”). Other alternatives not explicitly stated would be slight decrease in antibiotic penetration/functionality (could have large difference in selective pressure) or fitness of planktonic was already at a local optima (constrained to evolve farther in experimental setup). Mention of these kinds of concerns would be appropriate.

Thank you for this comment. To clarify, we do not conclude that fitness rather than antibiotic resistance was selected in the biofilm; rather, in our bead model, selection acted on mutations that balance growth and maintenance in the biofilm life cycle as well as resistance to CIP. Figure 3 demonstrates that mutants evolved in the biofilm treatment are relatively more fit than those from the planktonic phase, but are less resistant likely because the biofilm indeed protects cells from antibiotic penetration or activity (perhaps owing to slower growth). While these points were in the original manuscript, we have emphasized them and referenced Harrison et al., 2017 and in the summarizing Figure 5. We thank the reviewers for the reminder to emphasize the varying selective pressures of our model of the biofilm life cycle, which could explain the counterselection against the resistance mutations of higher effect selected in planktonic conditions (Results and Discussion section).

2) In the clinical relevance section, the choice to show Figure 4 as a heat map for fold increase of susceptibilities assumes that antibiotic concentrations (MICs) are meaningful on a linear scale. This assumption should be stated.

If we understand this point, the heatmap may not define clinically significant changes in susceptibility. We acknowledge these changes may not meet clinical breakpoints for altered resistance but denote two-fold, four-fold, or greater changes in susceptibility (subsection “Evolutionary consequences of acquiring resistance”).

3) Many of the conclusions of this study rely on the calculation (Supplementary file 3) that all possible mutations were sampled at the same frequency by each population during the course of the evolution experiments and therefore, any differences in evolutionary outcome is due to selection (Results and Discussion section and elsewhere). However, the calculations underlying this assumption rely on two key assumptions that where not experimentally tested here and have been refuted in other scenarios: that mutations occur at a fixed rate and that mutations occur with a uniform distribution at all genomic sites.

We address this point in response to #4 below.

4) One reviewer stated this another way, with references, stating that rates are known to vary with environmental/lifestyle differences in bacteria and other organisms (reviewed Fitzgerald, 2019). Specifically, mutation rates are increased dramatically by CIP and other quinolone antibiotics (Cirz 2005; Kohanski, 2010; Pribis, 2019) and may be different between planktonic and biofilm cultures (Boles, 2004, 2008). Additionally, the intrinsic physical protection of biofilm growth may mean that the mutation rate induction by CIP is different between growth conditions. Without direct measurement of mutation rates in the relevant growth conditions (CIP doses, biofilm, planktonic), it is impossible to negate the possibility that mutation rates differ greatly between treatments and that these differences could contribute substantially to the evolved outcomes.Thus, either mutation rates in unselected genes should be measured experimentally or the assumptions mentioned each time they are invoked and the conclusions toned down. As an example, the Results and Discussion section, the observation of rapid fixation of the adeN alleles and not the gyrA and parC alleles does not necessarily "indicate" that they confer higher resistance. Resistance-conferring adeN and related mutations could simply occur more frequently (higher local mutation rate, a larger target size, and/or have fewer selective constraints for other gene functions) compared to the topoisomerase mutants that emerge at later timepoints/higher [CIP]. The interpretation/conclusion needs to be softened or additional experiments (competition between adeN and gyrA/parC at low [CIP]) are needed to support it.

We appreciate these important points and are glad to provide more nuance to the argument that lifestyle per se produces different dynamics and targets of resistance evolution, and that these differences cannot be explained by mutation bias. The essence of Points 3 and 4 is that the mutation rate itself differs in environments containing CIP, that cells in biofilm experience lower CIP concentrations, and thus mutation availability differs between these environments to an extent substantial enough to cause mutations in qualitatively different genes to reach high frequency or fix. We believe this is very unlikely, for the following reasons, which we also now summarize in the Results and Discussion section in the manuscript.

1) All evidence from theory [reviewed in (Desai and Fisher, 2007; Cooper, 2018)] states that populations of bacteria with low, wild-type mutation rates that are maintained at these sizes are not limited by mutation availability, so an increased mutation rate in either condition is unlikely to lead to wholesale changes in the available selection targets. To our best estimate, the effective population sizes in both treatments are 10^7^ or greater throughout the experiments (Subsection “Experimental evolution”), which means than ~98% of all base-pair substitution mutations in the *A. baumannii* genome have occurred by day 12 in these experiments, and most several times over, assuming the wild-type mutation rate (subsection “Evolutionary dynamics under CIP treatment”, revised Supplementary file 1). [Rates for mobile elements like IS may be even higher and ubiquitous]. While increasing the mutation rate might increase the likelihood of molecular parallelism between populations experiencing the same selective pressures, it would not explain for example the absence of *gyrA* mutations or the enrichment of efflux-regulating mutations in the biofilm populations, because these mutations were certainly available for selection to act upon. We do acknowledge that under an increased mutation rate, early-arising or more probable beneficial mutations could sweep and limit invasion of selectively equivalent mutations in different genes (Subsection “Experimental evolution”). To summarize: large populations propagated for ~80 generations will gain access to nearly all mutations, strong selection like an antibiotic will enrich for the most adaptive ones, and increasing mutation rate will likely only accelerate this process.

2) The remarkably high levels of molecular parallelism observed among replicates of the same lifestyle (biofilm or planktonic), even among lines propagated at constant sub-MIC or increasing CIP, provides clear additional evidence that the mutation rate was not limiting in either environment because each population independently acquired equivalent mutations. Again, the differences in the targets between environments demonstrates that selection favors different traits, with the high-frequency mutations representing the most fit and not simply the most available.

3) While evidence that fluoroquinolones can increase mutation rate is fairly robust, the actual magnitude of this effect is unclear, ranging from a 4-fold increase in the most stringent study that quantified mutation rates and spectra over thousands of generations of evolution in the near-absence of selection, using whole-genome-sequencing (Long et al., 2016), to ~20-fold or more in a study shared by the referees using many reporter genes and treatments (Pribis et al., 2019). Importantly, neither increase is of the magnitude likely to explain completely different evolved mutations, since the selected mutations inevitably occurred in both treatments at the wild-type rate. Further, neither of these studies indicate that CIP changes the genomewide spectra of mutations except for inducing prophage in *E. coli* (not relevant here), and certainly not in ways that could lead to completely different mutated genes reaching high frequency in the face of extremely strong selection.

Because of multiple, detailed studies about the rates and spectra of mutations in bacteria (including some of our own, e.g. (Lynch et al., 2016; Dillon et al., 2017), including several in the presence of fluoroquinolones, we do not believe further experimentation is justified. We are grateful for the opportunity to clarify how replicate evolution experiments involving large populations under strong selection almost always select for the most fit mutations in the selected environment, provided mutation supply is not limiting as we have shown [Table S1, (Cooper, 2018)].

5) The lack of preadaptation to biofilm growth could be a major confounding factor in the comparison of adaptive paths taken in the two growth conditions. The data presented in Figure 2—figure supplement 3 suggest that the biofilm cultures were under additional selective pressure to adapt to biofilm growth and, therefore, a mutation that improved both biofilm formation and CIP resistance (such as adeL) would be highly preferred to one that only improved CIP resistance.It is possible that an alternative experimental design involving preadaptation to the experimental growth condition (planktonic or biofilm) prior to CIP exposure would yield quite different adaptive trajectories, helping to separate the selective pressure of the antibiotic from that of the growth condition. In the absence of such additional experimentation, this possibility should be explicitly mentioned.

We apologize for the confusion. The lack of fixed mutations in any of the three biofilm or planktonic populations after 12 days of evolution in the absence of antibiotics (two replicates are shown in Panels C and D of Figure 2) suggest (1) selection in the absence of antibiotic does not act very strongly on particular mutations, (2) both biofilm and planktonic populations were similarly adapted to our experimental medium and conditions, and (3) that the 10 days of preadaptation in our experimental drug-free environment, which selected three mutations, (Supplementary file 2) were beneficial in both environments. Therefore, the mutations reaching high frequencies in the presence of antibiotic were selected by both the lifestyle and the antibiotic pressure, but not simply lifestyle. We rephrased the third paragraph of subsection “Evolutionary dynamics under CIP treatment” for clarification. We agree with the reviewer that mutations that improve both biofilm formation and CIP resistance will be selected (Results and Discussion section).

6) In Figure 4, please add some details about the qualities of these antibiotics. On the Y axis, do they cluster with targets at all? More explanation of the targets of each antibiotics is needed, because not all readers will know the targets.

This has been clarified in the figure and in the legend.